# Dynamics of Lactic Acid Bacteria Dominance in Sour Bamboo Shoot Fermentation: Roles of Interspecies Interactions and Organic Acid Stress

**DOI:** 10.3390/foods14203481

**Published:** 2025-10-12

**Authors:** Xinxin Zhang, Changfeng Zhang, Menglian Gong, Pao Li, Hui Tang, Liwen Jiang, Yang Liu

**Affiliations:** 1College of Food Science and Technology, Hunan Agricultural University, Changsha 410128, China; zxxhunauyjs@yeah.net (X.Z.); zhcfhnnd@163.com (C.Z.); gongml88@163.com (M.G.); lipao@mail.nankai.edu.cn (P.L.); 2Guangdong Provincial Key Laboratory of Utilization and Conservation of Food and Medicinal Resources in Northern Region, Shaoguan University, Shaoguan 512005, China; lokeytang@163.com

**Keywords:** sour bamboo shoots, bacterial community succession, lactic acid bacteria, organic acids

## Abstract

To elucidate the mechanisms underlying the formation of the bacterial community predominantly composed of lactic acid bacteria (LAB) in sour bamboo shoots, the dynamics of the bacterial community structure and the influence of organic acids on the growth of dominant species were investigated. The results showed that the dominant bacteria in sour bamboo shoots changed from *Enterobacteriaceae* at first to LAB after fermentation. Correlation analysis revealed that organic acids, especially lactic acid and acetic acid, had great effects on bacteria. Even low concentrations of organic acids had negative effects on the growth of *Enterobacter asburiae*, while LAB exhibited remarkable tolerance to 8 g/L of organic acids. Notably, *Levilactobacillus spicheri* displayed a normal growth rate during incubation in medium containing 4 g/L of malic acid for 48 h. This study clarified the regulation of organic acids on bacterial community formation and provided a new understanding for bacterial control in traditional fermented foods.

## 1. Introduction

Fermentation is one of the time-honored food processing methods. Studying the microbial community structure and metabolism in fermented foods is important for understanding traditional fermentation processes. Previous studies have shown that the succession of microbial communities was affected by biological and abiotic factors, such as raw materials, environmental conditions, and microbial interactions. Raw material characteristics determine the initial colonization of microorganisms. Song et al. [1] reported that only cabbage- and garlic-derived microorganisms can successfully ferment kimchi. Environmental parameters affecting microbial composition include temperature, acidity, and salt concentration. For example, in non-post-fermented Shuidouchi, microbial diversity increases at moderate temperatures, *Bacillus* and *Aneurinibacillus* increase, and fungi decrease at high temperatures [2]. Pretreatment of fresh corn forage with organic acids in a certain proportion could reduce the abundance of undesirable microorganisms such as *Klebsiella* and *Paenibacillus* in whole-plant corn silage [3]. Under a low salinity environment, *Cardiobacteria* and *Ruminococcus* increased in Spanish-style green table olives, which promoted metamorphism [4]. Further, the interspecific interactions during fermentation also affect both microbial communities and fermented products. Liu et al. reported that amino acids and nucleosides secreted by *Saccharomyces cerevisiae* in Maotai had a positive effect on *Limosilactobacillus panis* (*Lm. panis*), while the produced ethanol had a negative effect on *Lm. panis* [5].

Sour bamboo shoots are a fermented food produced in southern China, which is also a common condiment in southern dishes. Usually, bamboo shoots (mainly *Dendrocalamus latiflorus*) are cut into pieces or chopped, then sealed and soaked in salt-free water for 30 days or more [6]. Although fresh bamboo shoots harbor diverse bacteria such as *Enterobacter*, *Raoultella*, *Oxyphotobacteria*, and *Enterococcus* [6,7], LAB becomes the dominant bacteria in sour bamboo shoots after a period of fermentation [7,8,9]. These LAB are considered key bacteria related to the flavor of sour bamboo shoots. In addition to the special aroma provided by volatile substances such as p-cresol and octanoic acid [10], the unique sourness related to organic acids is also an important flavor characteristic of sour bamboo shoots [11,12,13]. Previous studies have indicated that changes in organic acids (such as malic acid and succinic acid) during the fermentation of sour bamboo shoots are closely related to the metabolism of flavor compounds [9]. Guan et al. [7] also found that lactic acid and acetic acid are positively correlated with the microbial community of sour bamboo shoots in the mid and late stages of fermentation. However, the influence of organic acids on bacterial community succession in fermented bamboo shoots remains unclear.

The changes in microbial communities during sour bamboo fermentation revealed by high-throughput sequencing technologies could document the succession from non-LAB to LAB. However, the intrinsic mechanisms driving this succession have not been elucidated, as the DNA obtained by high-throughput sequencing represents the total bacterial genetic material in the sample and cannot distinguish between live and dead cells. It is necessary to reveal the growth dynamics of key bacterial groups (such as *Levilactobacillus spicheri* and *Enterobacter asburiae*). Further, the influence of organic acids on the proliferation of key bacterial groups could help reveal how organic acids shape the community structure through inhibition and selection. This study aims to reveal how organic acids shape the community structure through inhibition and selection and to provide a theoretical basis for a comprehensive understanding of the microbial community succession in sour bamboo shoots and optimize the process of sour bamboo shoots.

## 2. Materials and Methods

### 2.1. Sample Preparation and Fermentation Process

Clay jars were purchased from a local market and used as fermentation containers. The jars were sterilized by flaming: a small volume of ethanol was sprayed into the inner surface and then ignited. After the flame was extinguished, the jars were cooled to room temperature for use. Fresh bamboo shoots harvested from Liuzhou, China, were used as the fermentation substrate. The shoots were peeled, sliced, then put into clay jars and immersed in sterile water at a ratio of 1:1.2. The jars were sealed with water at the edge.

Sampling was conducted aseptically on days 0, 1, 4, 7, 10, 15, 20, 25, and 30 of fermentation. For each sampling, the entire jar was transferred into a laminar flow, and then the external surface of the jar was wiped with 75% (*v*/*v*) ethanol and exposed to ultraviolet radiation for 15 min before opening. The sample was then thoroughly homogenized by shaking. The unfermented (day 0) control sample was handled immediately after collection in the same manner as the fermented sample, with sterile water added for analysis. Each sample was divided into three parts.

### 2.2. Analysis of Microbial Community

DNA extraction: Microbial biomass was collected by centrifuging 25 mL of fermentation brine. DNA was extracted using the Magnetic Soil and Stool DNA Kit (TIANGEN, Beijing, China); 16S rDNA sequencing: target region: V3-V4 hypervariable region of bacterial 16S rRNA gene. Primers: 341F (5′-CCTAYGGGRBGCASCAG-3′) and 806R (5′-GGACTACNNGGGTATCTAAT-3′) were purchased from Sangon Biotech (Shanghai, China). Qualified PCR products were evaluated using an Agilent 2100 Bioanalyzer (Agilent, Santa Clara, CA, USA) and Illumina library quantitative kits (Kapa Biosciences, Woburn, MA, USA), which were further pooled together and sequenced on an Illumina NovaSeq 6000 (Illumina, San Diego, CA, USA).

### 2.3. Analysis of Physicochemical Properties

The second part was used for the analysis of immediate physicochemical properties. pH was measured using a calibrated pH meter (Lei Ci, Shanghai, China) on each sample. Total acidity was determined by titration with NaOH, expressed as lactic acid equivalent (g/kg). The composition of organic acids was analyzed by High-Performance Liquid Chromatography (HPLC), which was performed according to the research of Tang et al. [12]. HPLC conditions: Instrument: Liquid Chromatograph (Shimadzu, Kyoto, Japan); Column: ZORBAX Eclipse XDB-C18 (4.6 × 250 mm, 5-μm, Agilent, Santa Clara, CA, USA); mobile phase: 0.1% phosphoric acid-methanol = 97.5 + 2.5 (volume ratio), isocratic elution at 0.8 mL/min; detection: UV detector at 210 nm; standards including oxalic acid (99.6%), acetic acid (99.8%), succinic acid (99.5%), and propionic acid (99.5%) were purchased from Macklin (Shanghai, China). Malic acid (98%), L-lactic acid (98%), and citric acid (99.5%) were obtained from Aladdin (Shanghai, China). L-lactic acid was detected as studies have shown that the type of lactic acid in many fermented foods is L-lactic acid [14,15,16].

The third part was stored at −80 °C for future analysis. These samples were freeze-dried, ground into a powder, and subsequently used for the determination of total sugar and reducing sugar content. Reducing sugar and total sugar: the 3,5-Dinitrosalicylic acid (DNS) colorimetric method [17] was used to determine the reducing sugar before hydrolysis and total sugar after acid hydrolysis.

### 2.4. Isolation and Functional Validation of Key Strains

#### 2.4.1. Strain Isolation

LAB: MRS agar. *Enterobacteriaceae*: Violet Red Bile Glucose agar (VRBGA). Both culture media were purchased from Huankai Bio-Technology Co., Ltd. (Guangzhou, China). Physiological and biochemical experiments were conducted, including the Calcium dissolution test, Gram staining, litmus milk test, catalase test, oxidase test, gelatin liquefaction test, methyl red test, and indole test. The physiological and biochemical identification of LAB referred to Öz et al. [18] and Shao et al. [19], and *Enterobacteriaceae* referred to Padovani et al. [20]

#### 2.4.2. Mono- and Co-Culture Experiments

Simulated fermentation: LAB and *Enterobacteriaceae* strains were inoculated individually (mono-culture) and in combination (co-culture at a ratio of 1:1) into sterile bamboo shoot medium. The sterilized bamboo shoot medium was prepared by homogenizing fresh bamboo shoots with distilled water at a ratio of 1:1.2 (*w*/*v*). The mixture was then autoclaved at 105 °C for 30 min in conical flasks before use. The initial inoculum for all cultures was adjusted to approximately 10^8^ CFU/mL by the McNeill turbidimetric method, and the inoculation amount was 0.1%. The cultures were incubated at 37 °C under static conditions. The pH and plate count of the culture broth were measured daily over 7 days by aseptically sampling the medium. All experiments were performed in triplicate using 250 mL sterile conical flasks containing 100 mL of medium.

#### 2.4.3. Organic Acid Regulation Test

Oxalic acid, malic acid, lactic acid, and acetic acid were selected for tolerance assessment. Stock solutions of each organic acid were prepared, filter-sterilized, and aseptically added to sterile MRS (for LAB) or LB (for *Enterobacteriaceae*) medium to achieve final concentrations of 0, 0.5, 2, 4, and 8 g/L. The isolated strain and static culture were inoculated at 37 °C. Bacterial growth was monitored by measuring the OD_600_ every 12 h for 48 h using a microplate reader (ReadMax 1900, Shangpu, Shanghai, China). All absorbance readings were blank-corrected against the sterile medium. All assays were performed in sterile 96-well plates with three biological replicates, each with three technical replicates.

### 2.5. Statistical Analysis

All measurements were performed with three replicates, and the results were averaged. Principal Coordinates Analysis (PCoA) among communities was analyzed by R software (version 4.0.3); prediction of microbial community function was conducted using PICRUSt2; Pearson correlation calculation was conducted using an online platform (https://www.omicstudio.cn/tool, accessed on 21 January 2025). Canoco5 (ter Braak and Šmilauer, 2012) was used for Redundancy Analysis (RDA).

## 3. Results

### 3.1. Microbial Community Diversity of Sour Bamboo Shoots

Alpha diversity analysis and beta diversity analysis were performed based on the ASV feature sequence and ASV abundance table obtained from 2.2. Alpha diversity’s core lies in the comprehensive assessment of species richness, distribution uniformity, and completeness of sequencing data [21]. Figure 1A lists the core analysis parameters, including the observed_otus, Shannon, Simpson, and chao1 (logarithmic normalization of raw data was adopted for plotting). As the fermentation process progressed, chao1 increased, indicating a rise in community richness. In contrast, Shannon decreased, reflecting a reduction in community diversity. Beta diversity refers to the variation in species composition among different environmental communities [22]. PCoA was employed to examine the differences and similarities among samples. PCoA1 and PCoA2 contributed 94.16% and 4.25% to the OTU data difference. As illustrated in Figure 1B, the samples from D4 to D30 were closely clustered, particularly D4 and D7, as well as D15 and D30, indicating similarities in their microbial structures. In contrast, the samples from D0 and D1 were distinctly separated from all other sample groups, suggesting that the microbial composition of these two groups significantly differs from the others.

### 3.2. Representative Bacteria of Different Fermentation Stages of Sour Bamboo Shoots

The classification of bacterial biomes across the six sample groups at both the phylum and genus levels is illustrated in Figure 2. At the phylum level (Figure 2A), *Firmicutes* (51.38%), *Proteobacteria* (39.67%), *Bacteroidota* (3.7%), and *Actinobacteriota* (2.68%) together account for approximately 97.4% of the bacterial composition in fresh bamboo shoots (Group D0). As the fermentation time was prolonged, the proportion of Firmicutes gradually increased and came to dominate, rising to over 90%. The proportion of Actinobacteria fell below 1% in the later fermentation stages.

As can be seen from Figure 2B, the microbial composition typically undergoes very active changes during the early stages of fermentation [23]. In the raw materials (D0), *Weissella* was the dominant bacterium (24.55%), which also included many miscellaneous bacteria. With the fermentation process, LAB gradually occupied the absolute advantage. On the 30th day, the relative abundance of LAB was more than 90%. Among LAB, *Weissella* was the dominant bacterium in the early stage of fermentation. The relative abundance of *Weissella* reached 78.49% in D1. This surge was likely attributed to its rapid reproduction and effective utilization of fermentation substrates during the early stages, similar to the transient dominance of *Weissella* observed in the initial phase of kimchi fermentation [24]. After 30 days of fermentation, the abundance of *Weissella* gradually decreased to 7.10%. *Lactococcus* showed a similar trend, with its maximum abundance value appearing at D4, which was 25.58%. At the middle and late stages of fermentation (D15 and D30), *Lactiplantibacillus* became the dominant genus, and its relative abundance remained above 40%. It is worth noting that *Raoultella*, *Escherichia-Shigella*, *Pantoea,* and *Klebsiella*, which are members of the *Enterobacteriaceae* family and classified as facultative anaerobes, exhibited high abundances at D0, with relative percentages of 11.27%, 4.22%, 7.73%, and 3.76%, respectively. This phenomenon may be attributed to either their inherent presence in the raw materials or initial environmental contamination. However, the sum of *Enterobacteriaceae* relative abundances continued to decrease to only 1.34% by D30. It showed that LAB and *Enterobacteriaceae* bacteria grew in opposite directions in sour bamboo shoots; their interaction needed further investigation.

On account of 16S rDNA high-throughput gene recognition of live and dead cells, plate counts of *Enterobacteriaceae* and LAB were performed. In order to understand the growth dynamics of key bacterial groups (LAB and *Enterobacteriaceae*), the live cell counts of them during fermentation were detected Appendix A. *Enterobacteriaceae* could not be detected after 4 days of fermentation, while LAB rapidly proliferated and stabilized above 3.5 × 10^6^ CFU/mL (the counts were expressed as the log of CFU per milliliter in the figure). It was similar to the diversity changes in *Enterobacteriaceae* and LAB in sour bamboo shoots.

### 3.3. Functional Potential of Microbial Community

KEGG Ortholog (KO) information corresponding to OTU was obtained by Greengenes ID corresponding to OTU. Abundance of each functional category was calculated according to information from the KEGG database (Kyoto Encyclopedia of Genes and Genomes, http://www.genome.jp/kegg/, accessed on 15 December 2024) and OTU abundance [25]. Figure 3 shows two levels of information of the KEGG metabolic pathway. As can be seen from Figure 3A, the metabolism abundance was the highest at KEGG level 1 compared to environmental information processing and genetic information processing, and there was no significant difference in the relative abundance of metabolism among the sample groups. It indicated that metabolism is an important function during sour bamboo shoots fermentation, which is similar to paocai [26]. As can be seen from Figure 3B, more metabolism functions were identified at KEGG level 2, including amino acid metabolism, carbohydrate metabolism, energy metabolism, cofactors and vitamin metabolism, and nucleotide metabolism, among others. As the fermentation progressed, carbohydrate metabolism became higher and energy metabolism became lower, while the relative abundance of other metabolism pathways did not change significantly. Combined with Figure 2, the increase in carbohydrate metabolism abundance in the community may be related to the proliferation of LAB.

### 3.4. Changes in Physicochemical Properties in the Fermentation Process of Sour Bamboo Shoots

As shown in Figure 4A, the final content of reducing and total sugars decreased during fermentation, consistent with Guan et al. [7] and Singhal et al. [27]. This may be due to the increasing abundance of carbohydrate metabolic functions in the community (Figure 3B). However, reducing sugar content increased on day 7, probably because the rate of hydrolysis of total sugar into reducing sugars surpassed the rate of bacterial consumption of reducing sugars, leading to a transient accumulation of reducing sugars. The pH value of sour bamboo shoots decreased rapidly to 3.84 after the 4th day of fermentation and then tended to be stable. Total acid accumulated continuously and reached 28.35 g/kg but decreased to 26.19 g/kg on day 30.

According to Figure 4B, the total concentration of these seven organic acids exhibited a nonlinear dynamic characteristic, initially rising and then declining. The continuous accumulation of organic acids in the sour bamboo shoot led to a decrease in pH and an increase in total acids. On day 0, the predominant organic acids in the fermentation brine were oxalic acid, malic acid, and citric acid, indicating their presence in the raw bamboo shoot materials. This finding is consistent with the results reported by other scholars [9,28]. As fermentation progressed, these three organic acids gradually increased, with malic acid peaking on the 20th day at 5.64 g/kg. Subsequently, the decrease in malic acid concentration was potentially due to its involvement in the synthesis of other substances as precursors. Some studies have demonstrated that low-molecular-weight organic acids play a crucial role in various physiological and metabolic processes in plants [29]. However, lactic acid and acetic acid became the primary organic acids in the late stage of fermentation of sour bamboo shoots, and their concentrations reached 1.33 g/kg and 1.18 g/kg, respectively, on day 4. The lactic acid concentration reached a maximum of 7.16 g/kg after 20 days of fermentation and maintained a relatively high level until day 30.

### 3.5. Association Analysis of Microbial Interaction Networks and Environmental Factors

Pearson and coor.test were used to calculate the correlations among microorganisms and distance matrix correlations between microorganisms and environmental factors, and Figure 5 and a correlation coefficient table Appendix A were obtained. In Figure 5A, the heat map on the right shows the correlation between microorganisms. It reveals that *Weissella* was positively correlated with *Enterococcus* and *Raoultella* and significantly negatively correlated with other LAB (r < −0.6) within the sour bamboo shoots system. There was a positive correlation among *Lactiplantibacillus*, *Lactococcus*, *Limosilobacillus*, *Lactobacillus,* and *Levilactobacillus* (r = 0.36–0.90). At the same time, these five LAB genera were negatively correlated (r = −0.70 to −0.28) with *Enterobacteriaceae*, a negative correlation between LAB and *Enterobacteriaceae* that has also been reported in Mahewu [30], fermented carrot juice [31], and kimchi [32]. From the network diagram on the left in Figure 5A, there are only 11 pairs of statistically significant relationships that exist between the two distance matrices, including 4 correlations with total acid and microbial genera, 2 with pH and microbial genera, 3 with total sugar and microbial genera, and 2 with reducing sugar and microbial genera. This result indicates that pH and total acids are important indicators affecting microorganisms [33,34]; the direct regulation of the acid–base environment on the bacterial structure was significantly stronger than that of the carbon source.

The coor.test revealed overall correlations between environmental factors and bacteria, but it could not resolve interactions or collinearity among environmental factors, which may lead to misjudgments of key drivers. To address this limitation, RDA was further employed to visualize the direction and strength of associations among pH, total acid, and bacteria. In Figure 5B, the obtuse angle between total acid and *Weissella* or *Enterobacteriaceae* indicated negative correlations. The rapid decrease in *Enterobacteriaceae* was also found when the pH of the fermented cereal beverage fell below 4.5 [30]. On the contrary, total acid and organic acids were positively correlated with *Lactiplantibacillus*, *Lactococcus*, *Limosilactobacillus*, *Levilactobacillus,* and *Lactobacillus*. The vectors of lactic acid and acetic acid exhibited the smallest angles with LAB, coupled with significantly longer projection lengths compared to other organic acids, suggesting that they had stronger positive effects on LAB distribution. It is speculated that lactic acid and acetic acid could inhibit miscellaneous bacteria and promote LAB propagation.

### 3.6. Isolation and Identification of Key Strains and Exploration of Interactions

#### 3.6.1. Isolation and Screening of Strains

To analyze the interactions between key strains, a total of 36 strains were isolated by MRS medium and VRBGA medium and the Gram staining method, among which 28 strains of suspected LAB and 8 strains of suspected *Enterobacteriaceae* were screened (Figure 6A). After physiological and biochemical analysis, ten strains of LAB and four strains of *Enterobacteriaceae* were screened (Figure 6B). They were identified as seven *Lactiplantibacillus plantarum*, one *Lactiplantibacillus pentosus*, two *Levilactobacillus spicheri*, two *Enterobacter ludwigii,* and two *Enterobacter asburiae,* respectively. Further, they were inoculated into sterilized bamboo shoots alone (Figure 6C), and their growth was measured by the plate counting method. Three strains, *Enterobacter asburiae* (*E. asburiae*), *Levilactobacillus spicheri* (*Lv. spicheri*), and *Lactiplantibacillus plantarum* (*Lp. plantarum*), exhibiting the best growth were further investigated (Figure 6D). Several studies have confirmed the safety of *Lv. spicheri* [35] and *Lp. plantarum* [36]. Name abbreviations were based on the proposal of S.D. Todorov et al. [37].

#### 3.6.2. pH Dynamics and Bacterial Growth in Mono-Culture and Co-Culture Systems

Mono-culture of and co-culture of *Lv. spicheri*, *Lp. plantarum* and *E. asburiae* were carried out in sterilized bamboo shoots for 7 days. As shown in Figure 6E, the pH value decreased at first and then increased in mono-culture of *E. asburiae*, and the system reached a weak alkaline state on the 7th day. The initial pH decrease may be related to the leakage of organic acid from raw materials and the consumption of reducing sugar by *E. asburiae* to produce acid [38,39,40]. The subsequent pH increase may be caused by *E. asburiae* metabolizing protein. The pH of the system fermented by *Lv. spicheri* and *Lp. plantarum* continuously decreased to 3.46; in particular, the pH of the system fermented by *Lp. plantarum* decreased fast to 3.36 in 1 day, indicating that *Lp. plantarum* had a stronger acid production capacity. The situation was similar to that of naturally fermented products [9,41].

As shown in Figure 6F, the number of colonies increased in the first 2 days and then decreased slowly when *E. asburiae* was cultured alone. However, when co-cultured with *Lv. spicheri* or *Lp. plantarum* for 2 days, the total colonies of *E. asburiae* dropped below the detection limit. This indicates that *E. asburiae* can proliferate independently in mono-culture, while the decrease in pH caused by acid production from LAB in co-culture likely inhibits *E. asburiae* growth [30]. Notably, LAB colony counts initially increased and then decreased in all conditions: *Lv. spicheri* reached its peak on day 2 in mono-culture. Under other conditions, both *Lv. spicheri* and *Lp. plantarum* reached their peak on the first day.. At the late stage of *Lv. spicheri* and *E.asburiae* co-culture, the proliferation of *Lv. spicheri* was lower than that of *Lv. spicheri* mono-culture, probably because *E. asburiae* consumed some nutrients in the bamboo shoot. In addition, the decline rate of *Lv. spicheri* was significantly slower than that of *Lp. plantarum* in both mono-culture and co-culture, suggesting that *Lv. spicheri* may possess stronger acid tolerance.

### 3.7. Analysis of Concentration-Dependent Effects of Organic Acids on Growth Inhibition and Promotion of Isolated Strains

Since high levels of oxalic acid, malic acid, lactic acid, and acetic acid were observed in sour bamboo shoots, their effect on the growth of strains was further investigated. Figure 7A,D,G show that the inhibition of *E. asburiae* by organic acids differs by type and concentration. Lactic acid and acetic acid exhibited the most potent antimicrobial activity, which led to a decrease of over 75% in biomass compared to the control group. Similarly, oxalic acid also displayed significant dose-responsive inhibition, with 2 g/L and 4 g/L concentrations reducing *E. asburiae* biomass by 64% and 75%, respectively, at 48 h. The 0.5 g/L malic acid treatment led to a marginal *E. asburiae* biomass increase relative to the control in the first 36 h, suggesting potential substrate utilization at lower concentrations. This effect shifted to inhibition with increasing concentrations (above 2 g/L).

As can be seen from Figure 7B,E,H, and J (biomass variation in *Lv. spicheri *) and Figure 7C,F,I,K (biomass variation in *Lp. plantarum *), there were differential regulations of LAB by organic acids. *Lp. plantarum* reached high absorbance faster than *Lv. spicheri*, indicating that *Lp. plantarum* proliferates faster, which may be the reason that *Lp. plantarum* colony count reached the maximum value faster during bamboo shoot fermentation (Figure 6E). Interestingly, low concentrations of organic acids had no inhibitory effect on *Lv. spicheri* and *Lp. plantarum*. After 48 h of fermentation in a 0.5 g/L organic acid fermentation system, the absorbance of *Lv. spicheri* increased by 14% to 21% compared with the organic acid-free control. Notably, the biomass of *Lv. spicheri* showed a certain increase in the presence of 4 g/L of malic acid. Similarly, 2 g/L of organic acid had no effect on *Lp. plantarum* proliferation only in the early stage of fermentation (within 12 h). The inhibitory activity of organic acids on *Lv. spicheri* and *Lp. plantarum* proliferation increased gradually with the increase in organic acid concentration; in particular, oxalic acid and lactic acid showed the greatest effect on *Lv. spicheri* and *Lp. plantarum*. The growth of *Lv. spicheri* and *Lp. plantarum* was effectively inhibited when the oxalic acid concentration exceeded 2 g/L. *Lv. spicheri* exhibited stronger tolerance to high concentrations (8 g/L) of lactic acid and oxalic acid than *Lp. plantarum*, with the biomass of *Lv. spicheri* decreased by 59% to 76%, while that of *Lp. plantarum* decreased by 84% to 96%. The high tolerance ability of *Lv. spicheri* to lactic acid might be one reason for its slower decrease during sour bamboo fermentation (Figure 6E).

## 4. Discussion

The structure of microbial communities is very important to the quality of fermented products. At the phylum level, Firmicutes were the predominant phylum in sour bamboo shoots, as well as other fermented vegetables [23,42]. At the genus level, it was found that the relative abundance of the *Enterobacteriaceae* family (*Raoultella*, *Escherichia-Shigella*, *Pantoea,* and *Klebsiella*) was the highest in the early stage of fermentation, and the fermentation system dominated by the LAB family (*Lactiplantibacillus*, *Lactococcus*, *Limosilactobacillus,* and so on) was formed in the late stage of fermentation (Figure 2). These observations were similar to those of a cross-regional study of China’s traditional sour bamboo shoot fermentation [8,9,12] and other traditional fermented pickles [43]. However, the 16S rDNA sequencing cannot distinguish between live and dead cells. The detected *Enterobacteriaceae* in the later stage may come from dead cell DNA, which potentially overestimates their viability. This was confirmed by the live cell count data of LAB and *Enterobacteriaceae*
Appendix A.

Obviously, the changes in bacterial community structure, especially the increase in relative abundance of *Lactiplantibacillus*, led to an increase in the abundance of carbohydrate metabolism functions (Figure 3) [44], which further alter the content of lactic acid and acetic acid in sour bamboo shoot (Figure 4). Among them, the production of organic acids is mainly due to the release and permeation of raw materials and the metabolism of microorganisms to carbohydrates [39,45]. The main organic acid changes in the system may be due to LAB metabolizing malic acid, citric acid, and carbohydrates to produce lactic acid and acetic acid [46,47]. In the present study, organic acid profiling focused exclusively on L-lactic acid, and the presence of D-lactic acid was worth further quantifying to evaluate the safety of fermented foods.

Furthermore, the correlation analysis (Figure 5) and in vitro validation experiment collectively suggest that this succession, LAB gradually becoming the dominant bacteria, was likely driven by organic acids (especially lactic acid and acetic acid) produced by the metabolism of LAB. The inhibition of LAB on *E. asburiae* in the co-culture experiment (Figure 6) was likely achieved by creating a low pH and organic acid environment, since organic acids such as lactic acid and acetic acid have been widely proven to be antibacterial substances against *Enterobacteriaceae* [48,49]. Further, the pure culture experiment (Figure 7) confirmed that even low concentrations of organic acids can strongly inhibit the growth of Enterobacteriaceae, which could explain why *Enterobacteriaceae* rapidly die out in natural fermentation as organic acids accumulate.

Differently, organic acids showed dual regulation on LAB (Figure 7). Promotion at low concentrations is more likely due to its ability to reduce environmental pH, creating a competitive advantage for acidophilic LAB. The inhibitory effect of high concentrations of organic acids may be due to the destruction of the integrity of the cell wall structure of LAB. This result was consistent with the similar effect of phenolic acids on LAB reported by Rodríguez et al. [50]. Interestingly, *Lv. spicheri* can still grow normally in 4 g/L of malic acid, which contrasts with most strains that are inhibited under high acidity. This suggests that *Lv. spicheri* may possess unique acid tolerance mechanisms (such as specific stress responses) that allow it to maintain dominance in the harsh environment of the mid-to-late stages of fermentation. *Lv. spicheri* may have potential as a starter culture. However, in vitro culture experiments cannot fully simulate the complex physicochemical microenvironment in natural fermentation. RT-qPCR of key bacterial species combined with macro-transcriptomics could be considered to reveal the functions of active bacterial communities during fermentation.

In summary, the microbial succession during sour bamboo shoot fermentation resulted from interspecies competition and organic acid-mediated stress. It was driven primarily by the dominance of LAB, which outcompeted early-stage *Enterobacteriaceae* through the production of organic acids that both lower pH and generate antimicrobial substances. These findings deepen the understanding of the mechanism of colony formation in traditional food fermentation processes and help to develop functional fermentation agents centered around *Lv. spicheri* to regulate the sour bamboo shoot fermentation process.

## 5. Conclusions

This study revealed the mechanisms through which LAB achieved dominance during sour bamboo shoot fermentation. The dominant bacteria changed from *Enterobacteriaceae* to LAB (such as *Lactiplantibacillus*, *Lactococcus*, and *Limosilobacillus*) after fermentation. As the fermentation progressed, total sugar and reducing sugar eventually decreased, and total acid increased. Both lactic acid and acetic acid concentrations rapidly exceed 1.00 g/kg, thus inhibiting the proliferation of bacteria, especially *Enterobacteriaceae*. Interestingly, the regulatory effects of organic acids on LAB exhibited significant variations depending on both acid concentration and bacterial strains. The growth of LAB was promoted in a low concentration (0.5 g/L) of organic acids, especially *Lv. spicheri*, which increased by 14–21% in 0.5 g/L of organic acids compared with the organic acid-free control group. In contrast, high concentrations of organic acids (8 g/L) inhibit the growth of LAB, and *Lv. spicheri* showed better tolerance than *L. p* to organic acid. Therefore, this study revealed the nonlinear regulation of organic acid during bacterial community succession in sour bamboo shoots, which provided theoretical support for the regulation of bacterial population in the traditional fermentation system.

## Figures and Tables

**Figure 1 foods-14-03481-f001:**
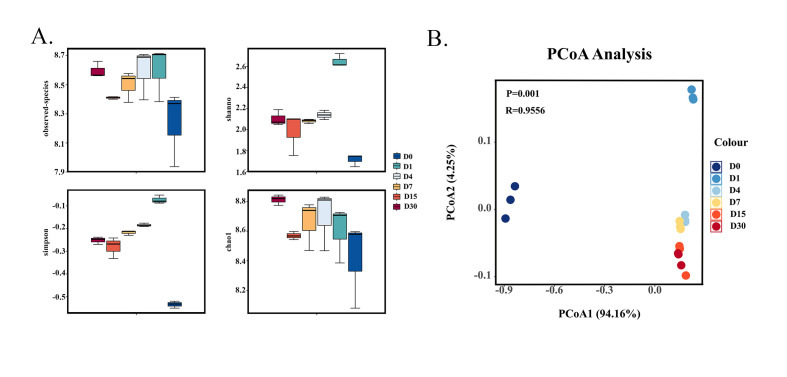
Alpha diversity (**A**) and beta diversity (**B**) of bacterial communities during sour bamboo shoot fermentation at the OTU level: The percentages on the horizontal and vertical axes indicate the degree of explanation of the sample differences by the first and second axes.

**Figure 2 foods-14-03481-f002:**
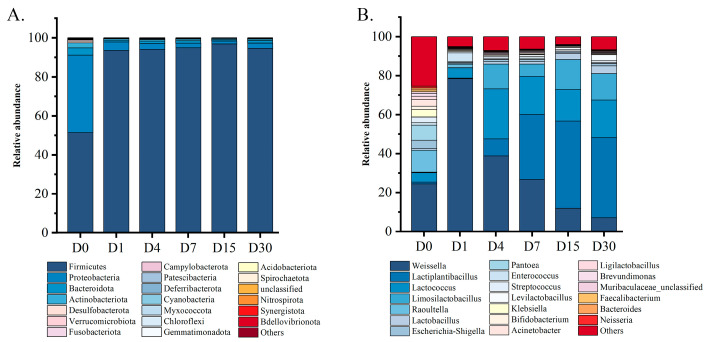
Changes in bacterial community structure at the phylum (**A**) and genus (**B**) levels during sour bamboo shoots fermentation. All measurements were performed with three replicates.

**Figure 3 foods-14-03481-f003:**
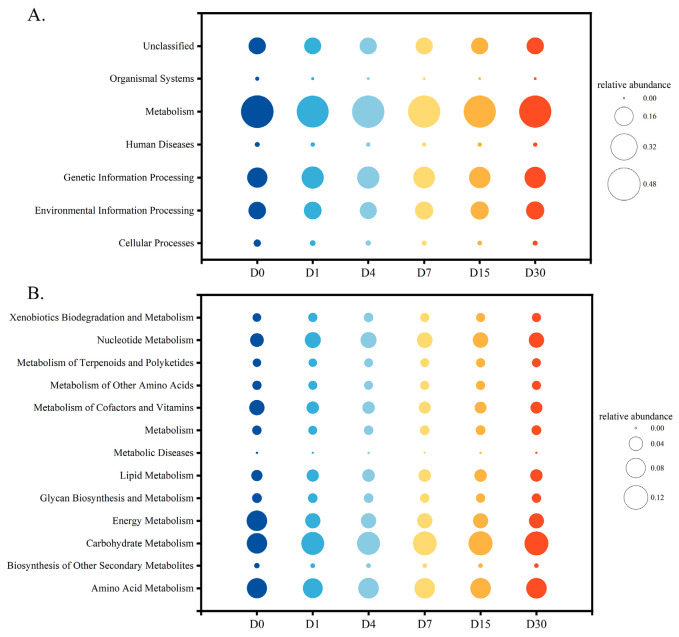
Predicted KEGG functional abundance of the sour bamboo shoot bacterial community based on KEGG level 1 (**A**) and KEGG level 2 (**B**). The size of bubbles corresponds to the relative abundance of functions, and the colors represent different functions.

**Figure 4 foods-14-03481-f004:**
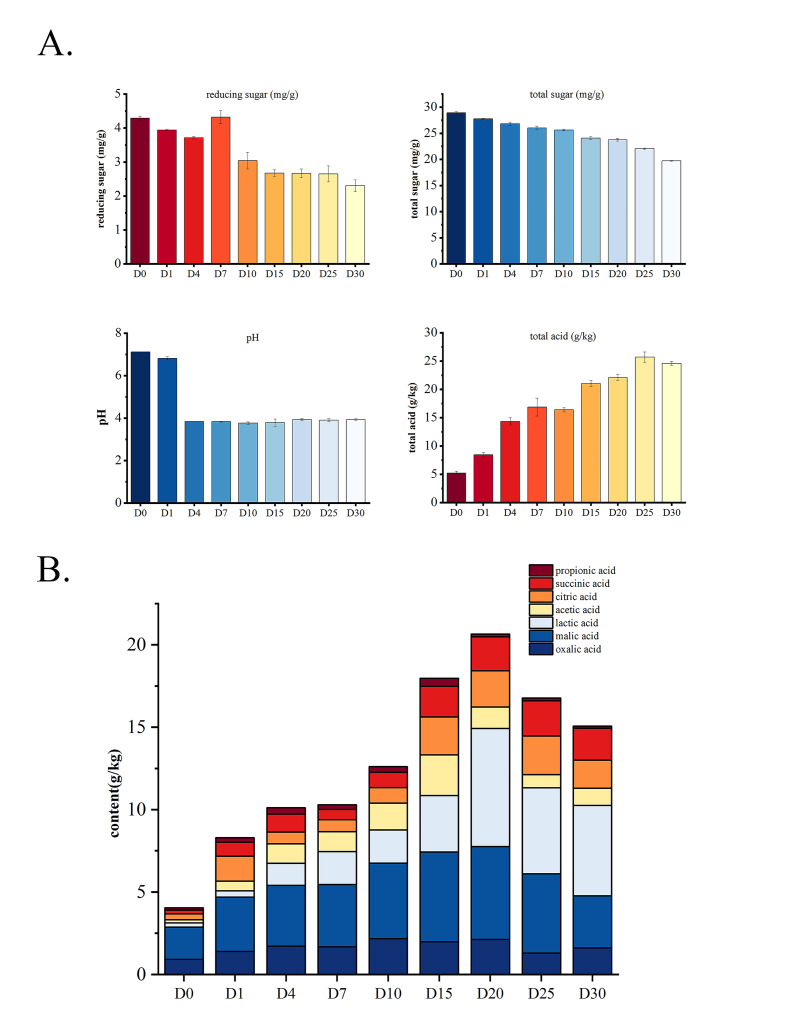
Changes in reducing sugar, total sugar, pH, total acid (**A**), and organic acids (**B**) during sour bamboo shoot fermentation.

**Figure 5 foods-14-03481-f005:**
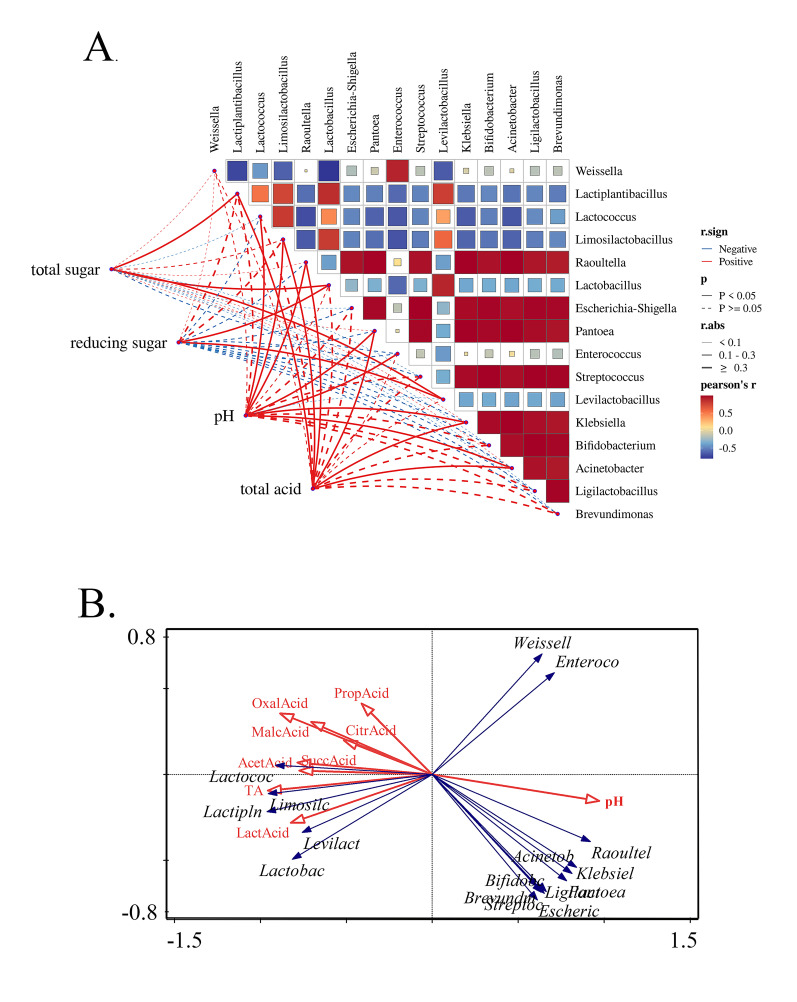
Association analysis of bacterial interaction networks and environmental factors (**A**) and RDA of genus and organic acids (**B**). Blue denotes negative correlation, red represents positive correlation, and |r| > 0.5 indicates strong correlation. Lines indicate significance; *p* < 0.05 indicates “r” is statistically significant.

**Figure 6 foods-14-03481-f006:**
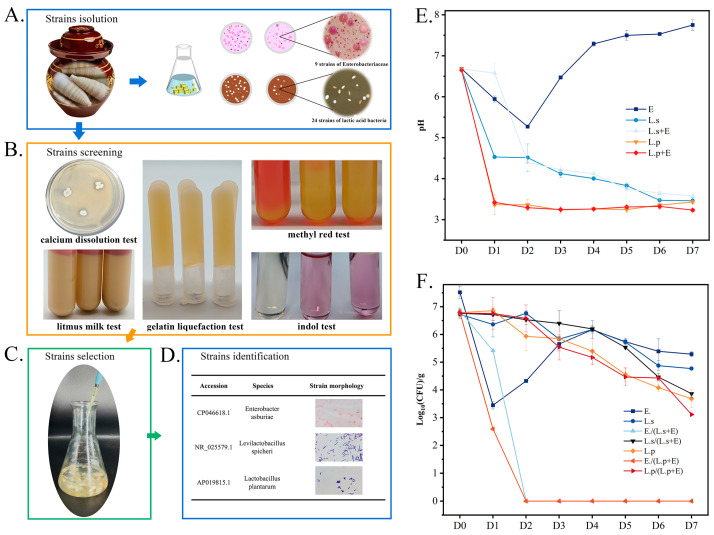
Screening of key strains (**A**–**D**) in sour bamboo shoots and changes in pH (**E**) and population (**F**) of strains in mono-culture and co-culture systems. Considering the size and aesthetic appeal of the figure, *Enterobacter asburiae*, *Levilactobacillus spicheri,* and *Lactiplantibacillus plantarum* are abbreviated as E., L.s, and L.p, respectively.

**Figure 7 foods-14-03481-f007:**
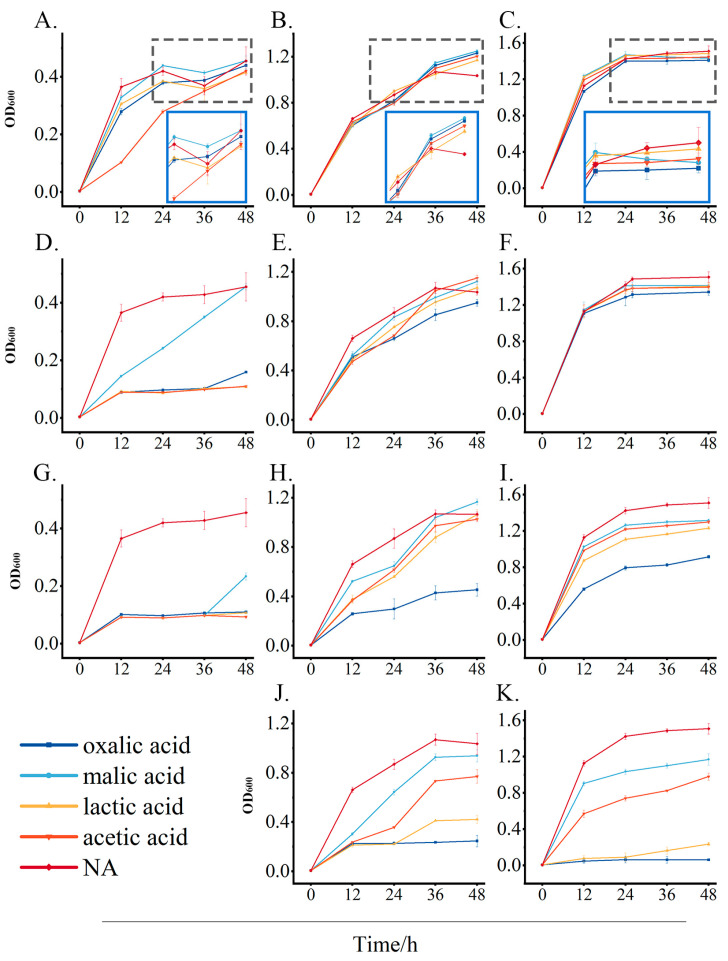
The effects of oxalic acid, malic acid, lactic acid, and acetic acid at concentrations of 0.5‰ (**A**–**C**), 2‰ (**D**–**F**), 4‰ (**G**–**I**), and 8‰ (**J**,**K**) on the growth of *Enterobacter asburiae* (**A**,**D**,**G**), *Levilactobacillus spicheri* (**B**,**E**,**H**,**J**), and *Lactiplantibacillus plantarum* (**C**,**F**,**I**,**K**). Biomass was expressed as OD_600_, and each measurement was repeated three times. The blue box was an enlarged representation of the black box (**A**–**C**).

## Data Availability

The original contributions presented in the study are included in the article/Appendix A. Further inquiries can be directed to the corresponding authors.

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
