# Peer review of "Dynamics of Lactic Acid Bacteria Dominance in Sour Bamboo Shoot Fermentation: Roles of Interspecies Interactions and Organic Acid Stress"

_foods, 2025, doi:10.3390/foods14203481_

Round 1
Reviewer 1 Report
Comments and Suggestions for Authors
The paper is interesting, however, in present form manuscript will need some additional attention, corrections, updates, clarifications. As research topic, the manuscript represents interesting contributions to the fermentation and production of fermented food products and authors have explored a topic that merit scientific attention.
Some of the difficulties are related to the fact that authors have not used correct templates (missing lines numbers) that is generating some inconvenience during review process.
In my opinion paper can be considered as carrying potential, however, it will need a major revision.
Some of the comments:
Have been tested concentration of lactic acid in general, or L and D-lactic acid separated? This can be important point related to safety.
Page 1: Correct to Levilactobacillus
Page 2: Correct to Limosilactobacillus panis. Please correct in other occasions and further abbreviate according to suggestions from 2023: Todorov SD, Baretto Penna AL, Venema K, Holzapfel WH, Chikindas ML. Recommendations for the use of standardized abbreviations for the former Lactobacillus genera, reclassified in the year 2020. Benef Microbes. 2023 Dec 12;15(1):1-4. doi: 10.1163/18762891-20230114. PMID: 38350480.
Page 2, Latin name for bamboo was already provided in the introduction. In my opinion it can be removed from material and methods section.
Page 2, section 2.1. What do you mean by "edgE"?
I have some doubts regarding applied procedure of fermentation and taking samples at different time periods. Are jars sterilized before applications? Any pretreatments? If the jars were sealed this is most probably to generate some microenvironmental conditions. However, if you are opening these jars every few days, this will be changing environmental conditions, maybe increasing oxidation, reducing volumes, etc., and this maybe will be not the best way to follow the fermentation process. Are authors proceeded with some precautions when jars were opened for taking samples? Are this done in controlled environmental conditions? Laminar? Anaerobic chamber? Please, maybe some details regarding this procedure need to be provided in the description.
If you have isolated DNA, then you may have results that will represent not only life, but dead cells as well. Have you applied additional steps in isolation procedures of DNA, so, can DNA be obtained only form life cells? Please, explain.
On 2.1. was mentioned that 2 parts of each samples were analyzed. One was for the physic-chemistry and other one for DNA. However, later in the work you mentioned microbiological evaluation. Please, in 2.1. needs to be corrected that in fact there were 3 parts.
Please, for all suppliers of material and equipment, follow standards of the Publisher and Journal. Name of the supplier needs to be accompanied by address, including city, state (in case of federal country) in abbreviated way and name of the country. However, on the following occasions, only name of the company will be sufficient. Please, try to use addresses of the headquarters and not that of the local distributors.
On Section 2.3. Please, correct "sample". Please, check entire manuscript for similar adjustments. This must be technical issue, kind of conflict between different alphabets loaded on the computer software.
Will be beneficial if authors can provide not only concentration of lactic acid, but information about proportions between L and D-lactic acids. Presence of D-lactic acid can be considered as safety issue and merit a bit more attention from the authors.
Please, provide more information regarding how test mentioned on 2.4.1. were performed or provide references regarding applied procedures. Use of term "etc." in 2.4.1. is not really appropriate, since all tests performed need to be mentioned.
Please, provide more details regarding procedures on 2.4.2. and 2.4.3. and references.
On page 10, change to Lactiplantibacillus plantarum; Lactiplantibacillus pentosus
Discussion need to be upgraded and get related to all experimental results obtained in current study.
Maybe authors can look for assistance from more experienced senior colleague that can help them in better structuring and discussing their results.
Author Response
Manuscript Number: foods-3907087
Dynamics of lactic acid bacteria dominance in sour bamboo shoot fermentation: Roles of interspecies interactions and organic acid stress
Article Type: Research Article
For Reviewer 1:
The paper is interesting, however, in present form manuscript will need some additional attention, corrections, updates, clarifications. As research topic, the manuscript represents interesting contributions to the fermentation and production of fermented food products and authors have explored a topic that merit scientific attention.
Some of the difficulties are related to the fact that authors have not used correct templates (missing lines numbers) that is generating some inconvenience during review process.
Response: We sincerely apologize for the negligence. Line numbers have been added.
In my opinion paper can be considered as carrying potential, however, it will need a major revision.
Some of the comments:
1. Have been tested concentration of lactic acid in general, or L and D-lactic acid separated? This can be important point related to safety.
Response: Thank you so much for your suggestion. In this experiment, the content of L-lactic acid in the sample was measured. Detailed explanations and references have been added on Page 3, Lines 113-115: “L-lactic acid was detected as some researches have shown that the type of lactic acid in many fermented foods was L-lactic acid [14-16]”.
2. Page 1: Correct to Levilactobacillus. Page 2: Correct to Limosilactobacillus panis. Please correct in other occasions and further abbreviate according to suggestions from 2023: Todorov SD, Baretto Penna AL, Venema K, Holzapfel WH, Chikindas ML. Recommendations for the use of standardised abbreviations for the former Lactobacillus genera, reclassified in the year 2020. Benef Microbes. 2023 Dec 12;15(1):1-4. doi: 10.1163/18762891-20230114. PMID: 38350480
Response: We sincerely apologize for the mistake. Thank you for providing a reference. In Lines 23, 48. “Levilactobacil-lus”, “Lactobacillus panis” have been changed into “Levilactobacillus” and “Limosilactobacillus panis”, respectively.The reference has been added on Page 7, Line 303.
3. Page 2, the Latin name for bamboo was already provided in the introduction. In my opinion, it can be removed from material and methods section.
Response: Thank you so much for your suggestion. The Latin name “Dendrocalamus latiflorus” of bamboo has been removed in Section 2.1 on Page 2, Line 82.
4. Page 2, section 2.1. What do you mean by "edgE"?
Response: Thank you so much for pointing out the mistake. This was a typographical error and has been corrected to “edge” on Page 2, Line 84.
5. I have some doubts regarding applied procedure of fermentation and taking samples at different time periods. Are jars sterilized before applications? Any pretreatments? If the jars were sealed this is most probably to generate some microenvironmental conditions. However, if you are opening these jars every few days, this will be changing environmental conditions, maybe increasing oxidation, reducing volumes, etc., and this maybe will be not the best way to follow the fermentation process. Are authors proceeded with some precautions when jars were opened for taking samples? Are this done in controlled environmental conditions? Laminar? Anaerobic chamber? Please, maybe some details regarding this procedure need to be provided in the description.
Response: We are sorry for the negligence. Before the experiment, jars were sterilized using flame sterilization. The specific details were added in on Page 2, Lines 79-81: “The jars were sterilized by flaming: a small volume of ethanol was sprayed into the inner surface, and then ignited. After the flame was extinguished, the jars were cool to room temperature for use.”; For the sampling method, the sampling procedures were conducted in a laminar flow, and the jars were opened for as short a time as possible. The specific details were added in on Page 2, Lines 86-88: “For each sampling, the entire jar was transferred into a laminar, then the external surface of the jar was wiped with 75% (v/v) ethanol and exposed to ultraviolet radiation for 15 minutes before opening.”
6. If you have isolated DNA, then you may have results that will represent not only life, but dead cells as well. Have you applied additional steps in isolation procedures of DNA, so, can DNA be obtained only form life cells? Please, explain.
Response: Thank you so much for your constructive suggestion. As the reviewer pointed out, live and dead cells cannot be differentiated by 16S rDNA sequencing, which is a known limitation of the technique. In the present study, additional plate counts for LAB and Enterobacteriaceae were also applied (Figure S1). Moreover, the description “To understand the growth dynamics of key bacterial groups (LAB and Enterobacteriaceae), the live cell counts of them during fermentation were detected (Figure S1). Enterobacteriaceae could not be detected after 4 days of fermentation, while LAB rapidly proliferated and stabilized above 6.5 (Total counts expressed as CFU per milliliter were log transformed). It was similar to diversity changes of Enterobacteriaceae and LAB in sour bamboo shoots.” were added on Page 5, Lines 205-210.
Figure S1 (Please see the attachment)
Figure S1. Plate count of Enterobacteriaceae and Lactic acid bacteria in sour bamboo shoots with different fermentation days. The horizontal axis represents time (days), and the vertical axis shows the log-transformed plate count data.
7. On 2.1. was mentioned that 2 parts of each sample were analyzed. One was for the physic-chemistry and other one for DNA. However, later in the work you mentioned microbiological evaluation. Please, in 2.1. needs to be corrected that in fact there were 3 parts.
Response: Thank you so much for your suggestion. The substance has been changed into “Each sample was divided into three parts” on Page 3, Line 91.
8. Please, for all suppliers of material and equipment, follow standards of the Publisher and Journal. Name of the supplier needs to be accompanied by address, including city, state (in case of federal country) in abbreviated way and name of the country. However, on the following occasions, only name of the company will be sufficient. Please, try to use addresses of the headquarters and not that of the local distributors.
Response: We are sorry for the negligence. The suppliers were added in Section 2.3 on Page 3, Lines 110-113: “Standards including oxalic acid (99.6%), acetic acid (99.8%), succinic acid (99.5%) and propionic acid (99.5%) were purchased from Macklin (Shanghai, China). Malic acid (98%), L-lactic acid (98%) and citric acid (99.5%) were obtained from Aladdin (Shanghai, China)”;
in Section 2.4.1 on Page 3, Lines 123-124: “Both culture media were purchased from Huankai Bio-Technology Co., Ltd (Guangzhou, China)”;
and in Section 2.4.3 on Page 4, Lines 146-147: “Bacterial growth was monitored by measuring the OD600 every 12 hours for 48 hours using a microplate reader (ReadMax 1900, Shangpu, China).”
9. Section 2.3. Please, correct "sample". Please, check entire manuscript for similar adjustments. This must be technical issue, kind of conflict between different alphabets loaded on the computer software.
Response: We sincerely apologize for the mistake. The word “sample” was corrected on Page 3, Line 105, and similar adjustments were applied.
10. Will be beneficial if authors can provide not only concentration of lactic acid, but information about proportions between L and D-lactic acids. Presence of D-lactic acid can be considered as safety issue and merit a bit more attention from the authors.
Response: Thank you so much for your suggestion. Same as question (1), we fully agree with the reviewer that distinguishing between L- and D-lactic acid is crucial for a comprehensive safety assessment of fermented foods. The references were added on Page 3, Lines 113-115: “L-lactic acid was detected as researches have shown that the type of lactic acid in many fermented foods was L-lactic acid [14-16]”. And the limitation was added in Discussion (on Pages 8-9, Lines 382-384): “In the present study, the organic acid profiling focused exclusively on L-lactic acid, and the presence of D-lactic acid was worth to further quantified to evaluate the safety of fermented foods”.
11. Please, provide more information regarding how test mentioned on 2.4.1. were performed or provide references regarding applied procedures. Use of term "etc." in 2.4.1. is not really appropriate, since all tests performed need to be mentioned. Please, provide more details regarding procedures on 2.4.2. and 2.4.3. and references.
Response: Thank you so much for your suggestion. The “etc.” was removed in Section 2.4.1 on Page 3, Line 127. More experimental parameters and references were added
in Sections 2.4.1 on Page 3, Lines 125-129: “Calcium dissolution test, Gram staining, litmus milk test, catalase test, oxidase test, gelatin liquefaction test, methyl red test, and indole test. The physiological and biochemical identification of LAB referred to Öz, Emel et al. [18] and Shao et al. [19], and Enterobacteriaceae was referred to Padovani et al. [20]”;
in Section 2.4.2 on Page 3, Lines 131-135, 139-140: “LAB and Enterobacteriaceae strains were inoculated individually (mono-culture) and in combination (co-culture at a ratio of 1:1) into sterile bamboo shoot medium. The sterilized bamboo shoot medium was prepared by homogenizing fresh bamboo shoots with distilled water at a ratio of 1:1.2 (w/v). The mixture was then autoclaved at 105°C for 30 minutes in conical flasks before use.”; “All experiments were performed in triplicate using 250 mL sterile conical flasks containing 100 mL of medium.”;
and in Section 2.4.3 on Page 4, Lines 142-149: “Oxalic acid, malic acid, lactic acid, and acetic acid were selected for tolerance assessment. Stock solutions of each organic acid were prepared, filter-sterilized, and aseptically added to sterile MRS (for LAB) or LB (for Enterobacteriaceae) medium to achieve final concentrations of 0, 0.5, 2, 4, and 8 g/L. Inoculate the isolated strain and static culture at 37°C. Bacterial growth was monitored by measuring the OD600 every 12 hours for 48 hours using a microplate reader (ReadMax 1900, Shangpu, China). All absorbance readings were blank-corrected against the sterile medium. All assays were performed in sterile 96-well plates with three biological replicates, each with three technical replicates.”
12. On page 10, change to Lactiplantibacillus plantarum; Lactiplantibacillus pentosus
Response: Thank you so much for your suggestion. The terms “Lactobacillus plantarum”, “Lactobacillus pentosus” have been changed into “Lactiplantibacillus plantarum” and “Lactiplantibacillus pentosus” in Section 3.6.1 on Page 7, Lines 296-297.
13. Discussion needs to be upgraded and get related to all experimental results obtained in current study.
Response: Thank you so much for your valuable suggestion. The improvements are as follows: (1) The revised Discussion articulates the relation among bacterial community succession (Figure 2), metabolism functions (Figure 3), organic acid accumulation (Figure 4), correlation analysis (Figure 5), co-culture (Figure 6), and organic acid regulation (Figure 7) experiments, proposing the conclusion of "organic acid-driven community succession". (on Pages 8-9, Lines 362-370, 375-382);
(2) The discussion on the limitations was added, and supplementary live cell count data (Figure S1) was provided as support. Additionally, feasible future research directions (e.g., integrated transcriptomics) were proposed (on Pages 8-9, Lines 370-374, 382-384,404-408);
(3) Discussed the significant tolerance of Levilactobacillus spicheri malic acid and explored its potential mechanism and value as a starter culture (on Page 9, Lines 400-404).
14. Maybe authors can look for assistance from more experienced senior colleague that can help them in better structuring and discussing their results.
Response: Thank you so much for your suggestion. We looked for assistance from professors at school by reviewing our manuscript and suggesting specific revisions. The revised parts were marked in red in the manuscript.

Reviewer 2 Report
Comments and Suggestions for Authors
I have reviewed the manuscript foods-3907087, titled ‘ Dynamics of lactic acid bacteria dominance in sour bamboo shoot fermentation: Roles of interspecies interactions and organic acid stress”, by Xinxin Zhang and co-authors.
The manuscript explores how lactic acid bacteria (LAB) become dominant in the fermentation of sour bamboo shoots. It tracks microbial community changes over a 30-day fermentation using 16S rRNA sequencing, alongside physicochemical measurements (pH, organic acids, sugars). The authors isolate key strains (Lactobacillus plantarum, Levilactobacillus spicheri, Enterobacter asburiae) and test their growth in mono- and co-cultures under different organic acid concentrations. This work contributes to understanding microbial succession and interspecies interactions in traditional fermentation, providing theoretical support for optimizing sour bamboo shoot production.
Objectives: The study’s objective is clear.
Novelty: The novelty of this study lies in combining community sequencing with isolation/functional validation of strains, addressing a gap in understanding interspecies dynamics in bamboo shoot fermentation. The authors should clarify in the Introduction how this study differs from previous research on sour bamboo shoots and other fermented vegetables.
Methodology:
Please, specify exact replicate numbers per experiment.
- More detail should be provided on the preparation of the "sterilized bamboo shoot medium" used in co-culture tests.
- Were controls (unfermented samples) maintained under identical storage conditions? Please clarify.
- 2.3. Analysis of Physicochemical Properties - Organic acid measurements by HPLC – please provide a reference for the method used.
- 2.4.1. Strain isolation – the authors report “Physiological and biochemical experiments were conducted, including Gram staining, catalase test, gelatin liquefaction test, oxidase test, methyl red test, etc.” please report which other tests took place and are not mentioned, but included in the etc.
- For the microbiology mediums (MRS agar, VRBGA), please provide the manufacturer.
- 2.4.3. Organic acid regulation test – please provide details about the spectrophotometer, model and manufacturer.
- 2.5. Statistical Analysis - For the statistical tools used statistics (R software, etc.) please provide the manufacturer of the respective software.
Results and Discussion:
The authors could discuss the practical implications of L. spicheri’s acid tolerance. Could it be used as a starter culture?
Figures: The legends could include more methodological detail regarding replicate numbers, statistical significance indicators, etc.
Author Response
Manuscript Number: foods-3907087
Dynamics of lactic acid bacteria dominance in sour bamboo shoot fermentation: Roles of interspecies interactions and organic acid stress
Article Type: Research Article
For Reviewer 2:
I have reviewed the manuscript foods-3907087, titled “Dynamics of lactic acid bacteria dominance in sour bamboo shoot fermentation: Roles of interspecies interactions and organic acid stress”, by Xinxin Zhang and co-authors.
The manuscript explores how lactic acid bacteria (LAB) become dominant in the fermentation of sour bamboo shoots. It tracks microbial community changes over a 30-day fermentation using 16S rRNA sequencing, alongside physicochemical measurements (pH, organic acids, sugars). The authors isolate key strains (Lactobacillus plantarum, Levilactobacillus spicheri, Enterobacter asburiae) and test their growth in mono- and co-cultures under different organic acid concentrations. This work contributes to understanding microbial succession and interspecies interactions in traditional fermentation, providing theoretical support for optimizing sour bamboo shoot production.
Objectives: The study’s objective is clear.
1. Novelty: The novelty of this study lies in combining community sequencing with isolation/functional validation of strains, addressing a gap in understanding interspecies dynamics in bamboo shoot fermentation. The authors should clarify in the Introduction how this study differs from previous research on sour bamboo shoots and other fermented vegetables.
Response: Thank you so much for your kind comments. The novelty was added in Introduction on Page 2, Lines 65-73 in revised manuscript: “Although the changes of microbial communities during sour bamboo fermentation revealed by high-throughput sequencing technologies could documented the succession from non-LAB to LAB, the intrinsic mechanisms driving this succession have not elucidated. Since the DNA obtained by high‑throughput sequencing represents the total bacterial genetic material in the sample and cannot distinguish between live and dead cells. It is necessary to the growth dynamics of key bacterial groups (such as Levilactobacillus spicheri and Enterobacter asburiae). Further, the influence of organic acids on the proliferation of key bacterial groups could help to reveal how organic acids shape the community structure through inhibition and selection.”.
Methodology:
2. Please, specify exact replicate numbers per experiment.
Response: We are sorry for the negligence. All measurements were performed with three replicates. The substance “All measurements were performed with three replicates and the results were averaged” was added on Page 4, Lines 151-152.
3. More detail should be provided on the preparation of the "sterilized bamboo shoot medium" used in co-culture tests.
Response: Thank you so much for your suggestion. The sterilized bamboo shoot medium was prepared by autoclaved at 105°C for 30 minutes in conical flasks prior to use. This description was added on Page 3, Lines 133-135.
4. Were controls (unfermented samples) maintained under identical storage conditions? Please clarify.
Response: Yes. The controls underwent the same storage conditions. The description “The unfermented (Day 0) control sample was handled immediately after collection in the same manner as the fermented sample, with sterile water added for analysis.” was added on Pages 2-3, Lines 89-90.
5. 2.3. Analysis of Physicochemical Properties - Organic acid measurements by HPLC – please provide a reference for the method used.
Response: Thank you so much for your suggestion. The substance “The composition of organic acids was analyzed by High-Performance Liquid Chromatography (HPLC), which was performed according to the research of Tang et al [12]” was added on Page 3, Lines 107-108.
6. 2.4.1. Strain isolation – the authors report “Physiological and biochemical experiments were conducted, including Gram staining, catalase test, gelatin liquefaction test, oxidase test, methyl red test, etc.” please report which other tests took place and are not mentioned, but included in the etc.
Response: Thank you so much for your suggestion. The “etc.” was removed and other tests were mentioned: including Calcium dissolution test, Gram staining, litmus milk test, catalase test, oxidase test, gelatin liquefaction test, methyl red test, and indole test. The description was added on Page 3, Lines 125-127.
7. For the microbiology mediums (MRS agar, VRBGA), please provide the manufacturer.
Response: We are sorry for the negligence. Manufacturer's information was revised on Page 3, Lines 123-124: “Both culture media were purchased from Huankai Bio-Technology Co., Ltd (Guang-zhou, China).”
8. 2.4.3. Organic acid regulation test – please provide details about the spectrophotometer, model and manufacturer.
Response: We are sorry for the negligence. The instrument's information was revised on Page 4, Lines 146-147: “Bacterial growth was monitored by measuring the OD600 every 12 hours for 48 hours using a microplate reader (ReadMax 1900, Shangpu, China).”
9. 2.5. Statistical Analysis - For the statistical tools used statistics (R software, etc.) please provide the manufacturer of the respective software.
Response: We are sorry for the negligence. The description “Pearson correlation calculation was conducted using online platform (https://www.omicstudio.cn/tool). Canoco5 (Microcomputer Power, America) was used for RDA” was added on Page 4, Lines 154-156.
Results and Discussion:
10. The authors could discuss the practical implications of L. spicheri’s acid tolerance. Could it be used as a starter culture?
Response: Thank you for your constructive suggestion. Levilactobacillus spicheri belongs to the genus Lactobacillus which was detected in the bacterial diversity on Day 0 (Fig. 2B), and only sterile water was added during processing, it indicated that Levilactobacillus spicheri was originated from raw bamboo shoots. Further, the safety of Levilactobacillus spicheri has been confirmed [35] (on Page7, Line 301). Consequently, Levilactobacillus spicheri may has potential as a starter culture. The discussion was added and on Page 9, Lines 400-404.
11. Figures: The legends could include more methodological detail regarding replicate numbers, statistical significance indicators, etc.
Response: Thank you for your suggestion. The figures captions were revised. The substance “Fig.1. Alpha diversity (A) and beta diversity (B) of bacterial communities during sour bamboo shoots fermentation at the OTU level.” was changed into “Figure 1. Alpha diversity (A) and beta diversity (B) of bacterial communities during sour bamboo shoots fermentation at the OTU level. The percentages on the horizontal and vertical axes indicated the degree of explanation of the sample differences by the first and second axes (B).”
The substance “Fig.7. Effects of oxalic acid, malic acid, lactic acid and acetic acid of different concentrations on the growth of Enterobacter asburiae (A), Levilactobacillus spicheri (B) and Lactobacillus plantarum (C).” was changed into “Figure 7. The effects of oxalic acid, malic acid, lactic acid, and acetic acid at concentrations of 0.5‰(A-C), 2‰(D-F), 4‰(G-I), and 8‰ (J, K) on the growth of Enterobacter asburiae (A, D, G), Levilactobacillus spicheri (B, E, H, J) and Lactiplantibacillus plantarum (C, F, I, K). Biomass was expressed as OD600, and each measurement was repeated 3 times.”
Round 2
Reviewer 1 Report
Comments and Suggestions for Authors
foods-3907087-peer-review-v2
Authors have improved the manuscript, and paper can be considered as appropriate to be suggested for publication. However, some points still need to be clarified and adjusted.
Please, for all suppliers of material and equipment, follow standards of the Publisher and Journal. Name of the supplier needs to be accompanied by address, including city, state (in case of federal country) in abbreviated way and name of the country. However, on the following occasions, only name of the company will be sufficient. Please, try to use addresses of the headquarters and not that of the local distributors.
On Ln48: When for the second time Limosilactobacillus panis was mentioned, it needs to be abbreviated to Lm. panis. Please, see DOI: 10.1163/18762891-20230114 for examples for abbreviations.
Ln93: The new introduced sentence do not make sense. Please check and correct.
Authors again provided revision without incorporating figures. Well, on this occasion, I have the previous version, but authors will need to be more careful when uploading the manuscripts to the editorial platform.
Author Response
foods-3907087-peer-review-v2
Authors have improved the manuscript, and paper can be considered as appropriate to be suggested for publication. However, some points still need to be clarified and adjusted.
1. Please, for all suppliers of material and equipment, follow standards of the Publisher and Journal. Name of the supplier needs to be accompanied by address, including city, state (in case of federal country) in abbreviated way and name of the country. However, on the following occasions, only name of the company will be sufficient. Please, try to use addresses of the headquarters and not that of the local distributors.
Response: We sincerely apologize for this oversight. The details were added in on Page 3, Lines 94-100: “DNA was extracted using the Magnetic Soil and Stool DNA Kit (TIANGEN DP712-02, Beijing, China). 16S rDNA sequencing: Target region: V3-V4 hypervariable region of bacterial 16S rRNA gene. Primers: 341F (5′-CCTAYGGGRBGCASCAG-3′) and 806R (5′-GGACTACNNGGGTATCTAAT-3′) were purchased from Sangon Biotech (Shanghai, China). Qualified PCR products were evaluated using an Agilent 2100 Bioanalyzer (Santa Clara, CA, USA) and Illumina library quantitative kits (Kapa Biosciences, Wo-burn, MA, USA), which were further pooled together and sequenced on an Illumina NovaSeq 6000 (San Diego, CA, USA).”
2. On Ln48: When for the second time Limosilactobacillus panis was mentioned, it needs to be abbreviated to Lm. panis. Please, see DOI: 10.1163/18762891-20230114IF: 3.1 Q2 for examples for abbreviations.
Response: Thank you so much for your suggestion. “Limosilactobacillus panis “has been abbreviated as “Lm. panis” on Page 2, Line 48.
3. Ln93: The new introduced sentence do not make sense. Please check and correct.
Response: Thank you so much for your suggestion. This sentence has been removed on Page 3, Line 93.
4. Authors again provided revision without incorporating figures. Well, on this occasion, I have the previous version, but authors will need to be more careful when uploading the manuscripts to the editorial platform.
Response: We sincerely apologize for this oversight. The new revised manuscript has been correctly uploaded to the editorial system.
Reviewer 2 Report
Comments and Suggestions for Authors
All previous comments and suggestions have been addressed, and the manuscript is now significantly improved.
Author Response
Thank you so much for your comments.